# The Fate of RPE Cells Following hESC-RPE Patch Transplantation in Haemorrhagic Wet AMD: Pigmentation, Extension of Pigmentation, Thickness of Transplant, Assessment for Proliferation and Visual Function—A 5 Year-Follow Up

**DOI:** 10.3390/diagnostics14101005

**Published:** 2024-05-13

**Authors:** Lyndon da Cruz, Taha Soomro, Odysseas Georgiadis, Britta Nommiste, Mandeep S. Sagoo, Peter Coffey

**Affiliations:** 1The London Project to Cure Blindness, ORBIT, Institute of Ophthalmology, University College London (UCL), London WC1E 6BT, UK; lyndon.dacruz1@nhs.net (L.d.C.); odgeorgiadis@gmail.com (O.G.); bnommiste@tenpoint-tx.com (B.N.); p.coffey@ucl.ac.uk (P.C.); 2National Institute of Health and Care Research Biomedical Research Centre at Moorfields Eye Hospital NHS Foundation Trust, Institute of Ophthalmology, University College London (UCL), London WC1E 6BT, UK; mandeep.sagoo1@nhs.net; 3Moorfields Eye Hospital NHS Foundation Trust, 162 City Road, London EC1V 2PD, UK; 4Wellcome/EPSRC Centre for Interventional & Surgical Sciences (WEISS), Charles Bell House, London W1W 7TY, UK; 5Center for Stem Cell Biology and Engineering, Neuroscience Research Institute, University of California, Santa Barbara, CA 93106, USA

**Keywords:** retinal pigment epithelium, neovascular age-related macular degeneration, submacular haemorrhage, stem cell therapy, human embryonic stem cells

## Abstract

(1) Background: We reviewed a stem cell-derived therapeutic strategy for advanced neovascular age-related macular degeneration (nAMD) using a human embryonic stem cell-derived retinal pigment epithelium (hESC-RPE) monolayer delivered on a coated, synthetic basement membrane (BM)—the patch—and assessed the presence and distribution of hESC-RPE over 5 years following transplantation, as well as functional outcomes. (2) Methods: Two subjects with acute vision loss due to sub-macular haemorrhage in advanced nAMD received the hESC-RPE patch. Systematic immunosuppression was used peri-operatively followed by local depot immunosuppression. The subjects were monitored for five years with observation of RPE patch pigmentation, extension beyond the patch boundary into surrounding retina, thickness of hESC-RPE and synthetic BM and review for migration and proliferation of hESC-RPE. Visual function was also assessed. (3) Results: The two study participants showed clear RPE characteristics of the patch, preservation of some retinal ultrastructure with signs of remodelling, fibrosis and thinning on optical coherence tomography over the 5-year period. For both participants, there was evidence of pigment extension beyond the patch continuing until 12 months post-operatively, which stabilised and was preserved until 5 years post-operatively. Measurement of hESC-RPE and BM thickness over time for both cases were consistent with predefined histological measurements of these two layers. There was no evidence of distant RPE migration or proliferation in either case beyond the monolayer. Sustained visual acuity improvement was apparent for 2 years in both subjects, with one subject maintaining the improvement for 5 years. Both subjects demonstrated initial improvement in fixation and microperimetry compared to baseline, at year 1, although only one maintained this at 4 years post-intervention. (4) Conclusions: hESC-RPE patches show evidence of continued pigmentation, with extension, to cover bare host basement membrane for up to 5 years post-implantation. There is evidence that this represents functional RPE on the patch and at the patch border where host RPE is absent. The measurements for thickness of hESC-RPE and BM suggest persistence of both layers at 5 years. No safety concerns were raised for the hypothetical risk of RPE migration, proliferation or tumour formation. Visual function also showed sustained improvement for 2 years in one subject and 5 years in the other subject.

## 1. Introduction

Treatment of retinal diseases using stem cell-based therapies [1] is a relatively new field of ophthalmic research. Stem cell-derived retinal pigment epithelium (RPE) has been the main cell type studied. Delivery of stem cell-derived RPE has centered on two main techniques: cells in suspension [2,3,4] or as a monolayer on a substrate, often termed a ‘patch’ [5,6,7]. Initial studies examining these technologies have shown evidence of visual recovery [4,5,6], which has been tempered by safety concerns in some cases [8,9,10].

Initial research has reviewed the safety and efficacy of these stem cell-derived RPE therapies [4,5,6]. Outcomes for 4-year follow up data are available for early RPE suspension transplantation studies [2], with up to 1 year data for human embryonic stem cell (hESC)-derived patch transplantation [5,6], and up to 4 years for induced pluripotent stem cell (iPSC) sources [7,11]. The lack of histology in these studies made it impossible to prove the survival of the transplanted RPE cells; however, in all these cases the preservation of sub-macular pigmentation at the site of transplantation and corresponding signal on OCT, along with measurable photoreceptor function and structure, was considered as indicative of RPE survival. From a safety aspect, there was no evidence of tumour formation, ocular inflammation, uncontrolled cellular proliferation or migration of transplanted RPE. Specifically, previous studies using stem cell-derived RPE in suspension or on a substrate have shown no evidence of cell proliferation beyond the vicinity of the RPE loss [5,12].

We present the 5-year follow up findings of RPE pigmentation, extension beyond the patch, thickness of hESC-RPE and basement membrane (BM), as well as assessment of RPE migration and proliferation in the transplanted hESC-derived RPE monolayers, and visual function in two subjects with large sub-macular haemorrhages secondary to neovascular AMD.

## 2. Materials and Methods

### 2.1. hESC-RPE-Basement Membrane (BM) Transplant

The hESC-RPE patch production has been previously described [5]. Briefly, RPE cells were derived from the SHEF-1.3 hESC line (Stem Cell Derivation Facility, Centre for Stem Cell Biology (CSCB), University of Sheffield, Sheffield, South Yorkshire, UK), according to GMP guidelines [13]. The cells were immobilised for transplantation as a polarised monolayer on a plasma-derived human vitronectin-coated polyester membrane (Sterlitech, Kent, Washington, DC, USA). The final implant had a surface area of 17 mm^2^ and comprised approximately 100,000 RPE cells.

### 2.2. Patients, Surgical Procedures and Ocular Oncology Assessments

Approval was granted from the UK Medicines and Health Products Regulatory Authority (MHRA), the Gene Therapy Advisory Committee (GTAC), the Moorfields Research Governance Committee and the London-West London & GTAC Research Ethics Committee. The study complied with Good Clinical Practice guidelines according to the European Clinical Trials Directive (Directive 2001 EU/20/EC) and the Declaration of Helsinki and had an independent External Data Monitoring Committee (E-DMC). Informed consent was obtained from each subject.

Two subjects, a 60-year-old female and an 84-year-old male, both with severe, acute vision loss in the study eye, due to sub-macular haemorrhage and RPE tear from neovascular AMD, were operated on in a phase I study (ClinicalTrials.gov NCT01691261). In both cases, conventional anti-VEGF treatment had previously failed. Subject 1 had received one intravitreal injection of ranibizumab, and subject 2 two monthly ranibizumab injections, before the sub-macular haemorrhage and an RPE tear occurred.

Standard 23-gauge pars plana vitrectomy was followed by induced macular detachment, clearance of sub-macular haemorrhage and insertion of the hESC-RPE-basement membrane (BM) implant into the subretinal space by a purpose-designed tool through a retinotomy. Silicone oil was used as tamponade and was subsequently removed with a second surgery. Both patients were followed up on for at least 5 years under a long-term follow-up trial protocol (ClinicalTrials.gov NCT03102138).

Ocular oncology team reviews were performed at week 2, week 4, week 8, 3 months, 4 months, 6 months, 9 months and 12 months post-operatively, with ocular ultrasound also performed at these visits. Liver ultrasound was performed at screening, 6 months and 12 months post-operatively. Regular blood tests reviewing liver function tests (LFTs) were also performed during the first year of the study. For the follow-up study where the subjects were reviewed at 18 months, 24 months, 36 months, 48 months and 60 months, the clinical investigator reviewed the subjects for evidence of any neoplastic formation. Ocular ultrasounds were also taken at these visits, and if there was any concern, the study subject could be referred urgently to the ocular oncology team.

### 2.3. Assessment of Patch Pigmentation and Retinal Pigment Epithelium Pigment Extension

Retinal pigmentation was assessed with colour fundus (CF) photography and spectral-domain optical coherence tomography (SD-OCT, on Heidelberg Spectralis) by clinical investigators (study ophthalmologists) and ocular oncologists as part of the study team.

### 2.4. Assessment of hESC RPE and BM Thickness

SD-OCT images were analysed using ImageJ2 software (2.15.0) [14] by clinical investigator TS. Each image was exported from the Heidelberg Heyex software (version 1.0) platform with the image having an associated 200 µm scale marker. The appropriate scale was then used for the OCT scan image in ImageJ2 by setting the scale to the defined distance of 200 µm on the image. Measurements of the thickness of the hESC-RPE and BM were then taken using the line tool. Five measurements were taken and repeated once per OCT B-scan, across areas of uniform hESC-RPE and BM.

### 2.5. Assessment of Visual Function; Best Corrected Visual Acuity (BCVA), Fixation and Microperimetry

Functional assessments of early diabetic retinopathy treatment study (ETDRS) BCVA, NIDEK-fixation and microperimetry were carried out by independent observers.

### 2.6. Statistical Analysis

Intraclass correlation coefficient (ICC) estimates of intra-rater agreement for measurement of hESC-RPE and BM thickness and their 95% confident intervals were calculated using Pingouin statistical package version 0.5.4 [15], based on a two-way mixed-effects model, for single rater absolute agreement. Jupyter notebook 6.5.2 [16] was used for the statistical package with python script (python 3.9.2) [17].

## 3. Results

### 3.1. Change in Patch Pigmentation

Subject 1 had patch pigmentation that was marked on the nasal parts while it was lighter on the rest of the patch (Figure 1A1). At 12 months, a central depigmented area developed (Figure 1A2), which enlarged over 5 years (Figure 2A2–A4), whilst temporal pigmentation persisted. The marked pigmentation on the nasal part of the patch was due to the patch sitting under the host RPE leading to a double layer of host and hESC-RPE (Figure 1B1,B2). The host RPE regressed over time, from implantation until year 5 (Figure 1A1–A4.)

Subject 2 had hypopigmented areas adjacent to the patch superiorly and inferiorly, as well on the inferotemporal aspect of the patch, immediately after implantation (Figure 2A1). These became less evident over the course of 5 years (Figure 2A2–A4). The inferotemporal hypopigmented area represented a persistent detachment of the hESC-RPE from the synthetic basement membrane, which is shown as a pigment epithelium detachment (PED) on the OCT (Figure 2B1,B2). There was also hyperpigmentation at the inferonasal portion which developed over the first 12 months (Figure 2A2), which gradually became heavier over 5 years (Figure 2A4).

### 3.2. RPE Pigmentation Extension

In both patients there was expansion of pigmentation outside the patch margins. It developed in a centrifugal manner, growing into the adjacent depigmented area around the rim of the patches. These findings were initially discussed for the first year in the study [5].

For subject 1, extension of the RPE progressed 1 month post-implantation (Figure 1A1) to its maximums spread at approximately 12 months post-implantation (Figure 1A2). There was regression of pigment extension at year 5 for subject 1(Figure 1A4).

For subject 2, extension of RPE was visible 1 month post-implantation (Figure 2A1) and progressed to its maximum extension at approximately 12 months post-implantation (Figure 2A2). Subject 2 had persistent preservation of RPE extension of the patch from 1 year onwards until the end of the study at year 5 (Figure 2A4).

The outward extension of RPE pigmentation in both cases was documented by optical coherence tomography (OCT) scans through the graft’s edges, showing a continuous outer retina, hyper-reflective line, consistent with the hESC-RPE extending from the grafted area and exteriorly to the patch (Figure 1C1,C2 and Figure 2C1,C2).

### 3.3. Retinal Ultrastructure

SD-OCT showed a continuous, hyper-reflective, double, outer retinal band, representing the two layers of the patch (hESC-RPE and synthetic BM) for both patients after implantation of the patch (Figure 3A1 and Figure 4A1). This persisted over at least some part of the patch for the 5-year follow-up period (Figure 3A2–A4 and Figure 4A2–A4). Both patients developed generalised retinal thinning over the 5-year period following patch transplantation.

Subject 1 had evidence of native RPE overlying hESC-RPE nasally, with associated cystic changes in the inner retina nasally at 3 months (Figure 3A1); this increased over time over 5 years with RPE hypertrophy (Figure 3A2–A4). The neurosensory retina had distinct, multilayer architecture after implantation, with an ellipsoid zone visible at 12 months (Figure 3A2); however, this became indistinct from 24 months onwards until year 5 (Figure 3A4).

Subject 2 developed a disorganised retinal layer architecture post-implantation. Associated with this, there were initial cystic changes and thinning at the centre of the patch, which stabilised at year 1 (Figure 4A1,A2). The ellipsoid zone was visible over some areas of the patch at 12 months and remained apparent at year 5 above the hESC-RPE (Figure 4A4), although in a smaller total area. The temporal retina continued to be thickened after 12 months, whilst the nasal retina had normal thickness over the 5-year period. Both patients developed epiretinal fibrosis, which was more prominent in subject 2.

### 3.4. hESC-RPE and BM Thickness

Predefined measurements of hESC RPE + BM are shown with histological ultrastructure (Figure 3A4*,A4§ and Figure 4A4*,A4§), with a combined thickness of 25 µm [5,18]. Average measurements of hESC-RPE and BM thickness over uniform areas, where there is no hESC-RPE detachment, or double host/hESC-RPE, were taken at 3 months, 12 months, 2 years and 5 years post-operatively. Measurements over these time points for both study participants remain within 2 µm of the predefined thickness of both layers. There is good intra-rater agreement for all measurements for all OCT B-scans (ICC 0.98 (CI 0.91, 1.0), Table 1).

### 3.5. RPE Migration and Proliferation

There was no evidence of distant donor RPE proliferation or migration beyond the hESC-RPE-BM bilayer for subject 1 (Figure 1A1–A4) and subject 2 (Figure 2A1–A4), apart from centrifugally to fill out areas of bare host basement membrane, at 5 years, as outlined previously. There was no evidence of neoplasm related to pigment extension by serial ocular ultrasound, OCT, colour fundus photography and review by ocular oncology for the first year of the study, and then by slit lamp examination, colour fundus photography and OCT assessment by the clinical investigator up to year 5.

### 3.6. Best Corrected Visual Acuity (BCVA)

BCVA for subject 1 and subject 2 had a significant visual improvement from baseline to year 2 (16 letter gain for subject 1 and 15 letter gain for subject 2, Table 2.). Subject 1 declined from year 2 until year 5 with BCVA below baseline at the end of the study (2 letters). Subject 2 maintained their BCVA improvement until year 5 (10 letter gain).

### 3.7. Fixation

At baseline, fixation with a red cross 2-degree target could not be assessed for subjects 1 or 2 (Figure 5A,B). Both subjects maintained stable fixation over the patch for the first year (Figure 5A,B at 12 months) with the small 2-degree fixation target remaining completely within the patch borders. Subsequently, subject 1 lost fixation over the patch (Figure 5A at 24 months), which continued to the last recorded period at 3 years (Figure 5A at 36 months). For subject 2, fixation with the target completely within the patch margins and over the center of the patch was maintained until the last record at 4 years (Figure 5 at 48 months).

### 3.8. Microperimetry

Both subjects had zero sensitivity recorded over the location of the submacular haemorrhage at baseline, with an improvement in retinal sensitivity at 1 year (Figure 5A,B at 12 months). Subject 1 (Figure 5A at 12 months) reached normal retinal sensitivity in the temporal half of the patch (18–20 dB) at 1 year, this deteriorated at 2 years (Figure 5A at 24 months), with a smaller area over the temporal half of the patch having reduced retinal sensitivity over the temporal half of the patch at 3 years (8–14 dB) (Figure 5A at 36 months). For subject 2, there were lower improvements in retinal sensitivity following patch insertion; however, areas of positive retinal sensitivity over the patch continued to the last recorded time-point at 4 years (Figure 5B at 48 months).

## 4. Discussion

In this paper, we present the 5-year observations of hESC-derived RPE patch pigmentation, pigment extension, OCT appearance of hESC-RPE and patch thickness, assessment for RPE migration, RPE proliferation and visual function post-transplantation. We report persistence of pigmentation, as a de facto indication of hESC-RPE cells, on the patch. Extension of pigmentation was limited to adjacent and contiguous spread over bare donor basement membrane locally. The extension of pigmentation occurred for up to 12 months, with no further extension after that period in either patient. There was some evidence of later regression. We also show an improvement in visual acuity for both subjects up to 2 years, and to 5 years for one subject.

Using imaging and clinical assessment by ocular oncologists and retinal specialists, both subjects in the study had no evidence of uncontrolled proliferation of RPE over the 5-year period (Figure 1A1–A4 and Figure 2A1–A4). Previous studies show that uncontrolled cell proliferation tends to occur within the first few months of transplantation [19,20,21]. Schwartz et al. have shown longer term safety up to 4 years using hESC-RPE cells in suspension [2]. Stem cell-derived RPE patch therapies have shown a similar safety up to 1 year [5,6], and up to 4 year for iPSC-derived RPE transplants [7,11]. Our study shows RPE patch safety, in terms of uncontrolled or distant RPE proliferation, for up to 5 years post-transplantation.

In terms of extension of the pigmented hESC-RPE monolayer, Mandai et al. have reported one case of iPSC-derived RPE patch transplantation to have a persistence and expansion of pigmentation combined with OCT segmentation to demonstrate survival of the patch and preservation of retinal ultrastructure [7,11]. Schwartz et al., in their earlier work on hESC-RPE cells in suspension, also demonstrated increased pigmentation in 72% of the host eyes [4], which they considered as thickening or reconstruction of the RPE. In our study, we showed that in both cases there was donor RPE pigment extension covering bare surrounding host basement membrane (Figure 1A2–A4 and Figure 2A2–A4) and stopping via likely contact inhibition with intact host RPE. This occurred within the first 12 months with some evidence of regression thereafter. This extension also gave a clear RPE-like signal on OCT, which was continuous with the hESC-RPE on the patch (Figure 1C1,C2 and Figure 2C1,C2). The fact that the movement of the pigmentation was centrifugal to the patch suggests that this represents hESC-RPE cell migration to cover the adjacent areas that are denuded of RPE due to disease and surgery. Further evidence to support preservation of the hESC-RPE for both subjects after 5 years is the measurement of hESC-RPE thickness, which continued to be within 2 µm of the predefined hESC-RPE and BM thickness of 25 µm [5,18].

Both our subjects maintained an improvement in functional tests to two years post-transplantation. We report a 29- and 16-letter VA gain for subject 1, and 21 and 15 for subject 2, at years 1 and 2, respectively. Subject 1′s BCVA gain returned to a 1-letter gain at year 4 and below baseline at year 5 to −8 letters. Subject 2 maintained a 10-letter VA gain at years 4 and 5. Both displayed improved fixation and microperimetry sensitivity at 1 year compared to baseline. Subject 1 had a subsequent decrease in both, whilst subject 2 maintained both fixation and microperimetry improvement until 4 years post-intervention. In the only other published patch transplantation case in nAMD, Mandai et al. [7,11] reported no VA improvement and 0 MP sensitivity. We acknowledge that the removal of the submacular haemorrhage may have contributed to the early post-operative visual recovery of our cases; however, this would not have been sustained for the long-term without a functioning RPE to support the overlying retina.

We present the presence of pigmentation as a de facto indication of hESC-RPE survival based on colocalisation of OCT signals that are of the same character as native RPE. Furthermore, the pigmented areas are relatively stable in appearance in consecutive reviews compared to the early post-transplantation imaging. We also present sustained BCVA improvements for 2 years for both patients, with subject 1 having a decline thereafter and subject 2 maintaining BCVA improvements until the end of the study. Fixation and microperimetry improvemens are evidenced for up to 1 year in subject 1 and up to 4 years in subject 2. Finally, we have previously documented, using structure/function correlation, that the areas of pigmentation characterised as persisting RPE were able to sustain visual function in terms of microperimetry and fixation and to sustain structural presence of cone photoreceptors. Despite this, without histology, it is not possible to confirm that the pigmented areas represent RPE. It is also not possible to be certain whether some of the areas of pigmentation, especially non-functional areas of hyperpigmentation, represent pigmented macrophages as part of a low-grade, subclinical, chronic rejection-related inflammation. Activation of those macrophages and migration into areas of cell loss could be a sign of late transplant rejection, though this could only be confirmed by future histological examination. Immune rejection has been reported as a significant cause of late failure after RPE transplantation and can occur from months to years after the operation [22]. It is more prevalent after allogeneic RPE grafts and has been associated with graft depigmentation, but also with macular oedema, leakage in fundus fluorescein angiography (FFA), fibrous encapsulation and recurrence of CNV [23,24]. None of the latter was observed in our cases over uniform hESC-RPE. Conversely, areas of pigmentary loss over time may represent loss of RPE due to chronic low-grade inflammation, loss of perfusion due to a disorganised and scarred choriocapillaris or progression of the primary disease.

## 5. Conclusions

For hESC-RPE patch transplantation to be useful as a future therapeutic strategy for macular diseases, it needs to be safe, survive and provide sustained improvement and visual stabilisation. We present our 5-year study findings which show no evidence of uncontrolled RPE proliferation or migration, only local, contiguous pigment extension which ceases at 12-months and preserved, normal hESC-RPE and BM thickness. We also demonstrate that pigment covers bare host basement membrane, which may represent donor RPE, which has been shown to occur in previous studies. Sustained improvement of visual acuity is also demonstrated for 2 years for the first patient and for 5 years for the second patient. More cases need to be assessed before we can draw firmer conclusions; however, these early cases provide important learning points for future work in this area.

## 6. Patents

2 patents are lodged by University College London (UCL) Business. The Patent Application No. PCT/GB2009/000917 (for the patch) and International Patent Application No. PCT/GB2011/051262 (for the surgical tool).

## Figures and Tables

**Figure 1 diagnostics-14-01005-f001:**
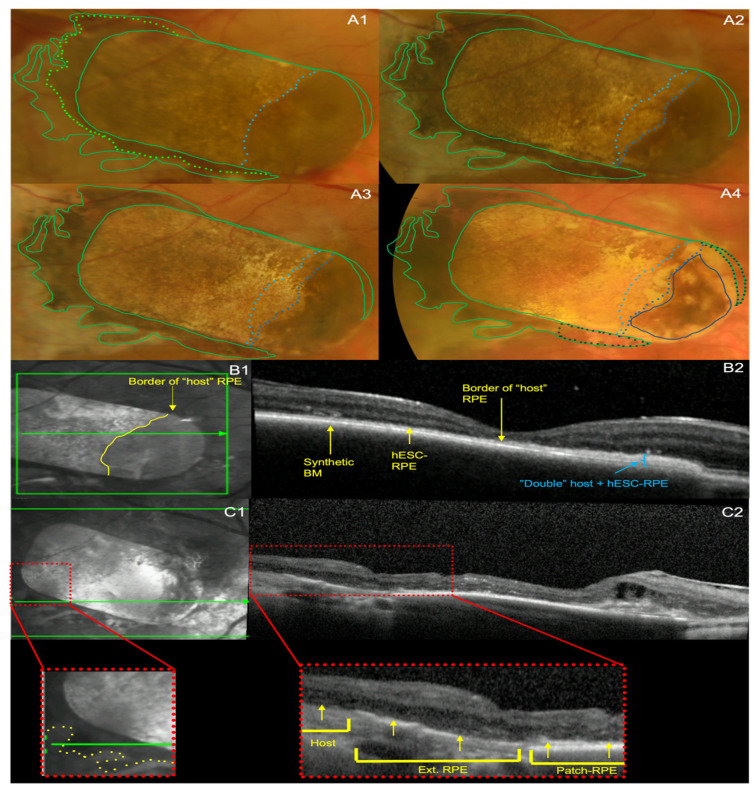
Changes in the pigmentation of the patch with time points and extension of pigment into the surrounding retinal area of subject 1. (**A1**–**A4**) Colour fundus photographs from (**A1**) month 1, (**A2**) month 12, (**A3**) year 2 and (**A4**) year 5 post-hESC-RPE patch implantation. The green sold line demonstrates the maximum extension of pigmentation outside the patch’s edges, which was reached at 12 months, this has been overlaid over all times points to highlight the progression and regression of RPE extension. The dotted lighter green line represents the extension of pigmentation at 1 month (**A1**). The dotted darker green line demonstrates areas of retraction of the extension of the RPE at year 5 (**A4**). The blue lines demonstrate the retraction of the darker (nasal) segment of the over-the-patch retina, deemed to be an area where host RPE overlay the implanted hESC-RPE (mentioned in text as ‘merged RPE’). Different shades of blue represent different time points for the borderline of this segment: the lighter shades correspond to earlier times and darker shades to later times (**B1**,**B2**), whilst the end stage of retraction of nasal patch RPE is represented by a solid dark blue outline. Near infrared photo and SD-OCT b-scan from the 2nd post-operative week of subject 1, showing the optimal positioning of the hESC-RPE on the synthetic BM patch (**B2** left half) and highlighting the abnormal thickness of the ‘merged’ (host + implanted) RPE (**B2** right half). Good retinal segmentation is demonstrated throughout the treated area (**B2**). (**C1**,**C2**) Near infrared and SD-OCT b-scan at 5 years post-operatively showing the outward extension of the implanted hESC-RPE. Higher magnification of the annotated parts (red dotted squares) highlighting the b-scan ‘cutting’ through a retinal area that includes sequentially a part of the patch, a part of the extended RPE (yellow line in **B1**) and a part of the host tissue. The RPE extension appears as a continuous outer hyper-reflective line starting at the hESC RPE (Patch-RPE) highlighted by yellow arrows and entering the adjacent host tissue as an extension of patch RPE (Ext. RPE), before stopping at areas of preserved host RPE (Host). Retinal segmentation appears well preserved over the patch.

**Figure 2 diagnostics-14-01005-f002:**
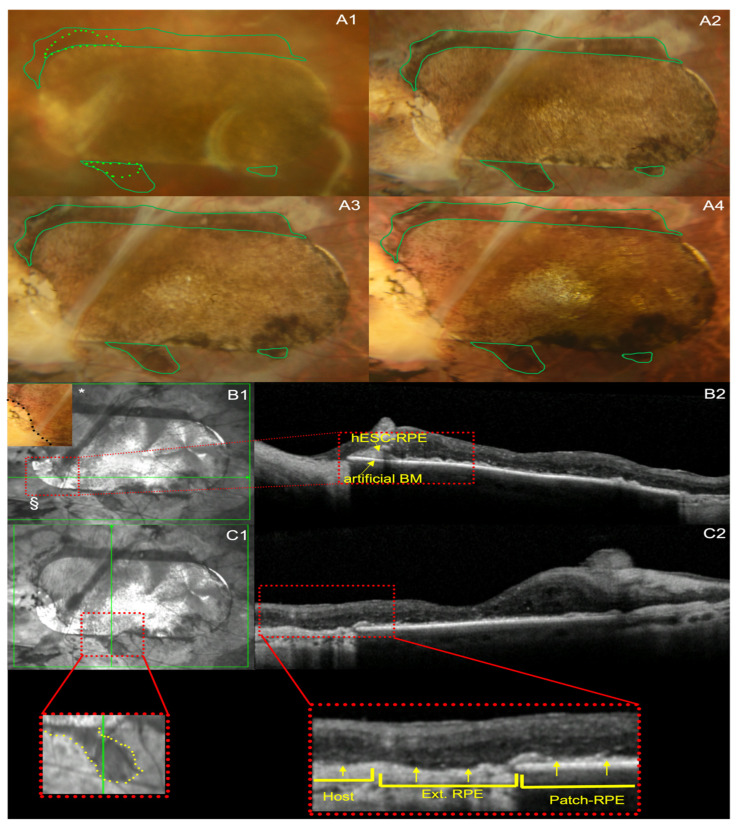
Changes in the pigmentation of the patch with time points and extension of pigment into the surrounding retinal area of subject 2. (**A1**–**A4**) Colour fundus photographs from (**A1**) month 1, (**A2**) month 12, (**A3**) year 2 and (**A4**) year 5 post-hESC-RPE patch implantation. The green sold line demonstrates the maximum extension of pigmentation outside the patch’s edges, which was reached at 12 months; this has been overlaid over all times points to highlight the progression and regression of RPE extension. The dotted lighter green line represents the extension of pigmentation at 1 month (**A1**). (**B1**,**B2**) Near infrared photo and SD-OCT b-scan from the 5-year post-operative period of subject 2, showing the stable position of the hESC-RPE-artificial BM implant, as well as a detachment of a segment of the hESC-RPE from the BM, at the inferior-nasal corner of the patch (**B1** * and § for colour and near infrared pictures, respectively), appearing as PED in the b-scan (**B2**). Retinal segmentation is preserved in the nasal half of the treated area. (**C1**,**C2**) Near infrared and SD-OCT b-scan at 4 years post-operatively, showing the outward extension of the implanted hESC-RPE. Higher magnification of the annotated parts (red dotted squares) highlighting the b-scan ‘cutting’ through a retinal area that includes sequentially a part of the patch, a part of the extended RPE (yellow dotted line in **C1**) and a part of the host tissue. The RPE extension appears as a continuous outer hyper-reflective line starting at the hESC patch RPE (Patch-RPE) highlighted by yellow arrows and entering the adjacent host tissue as an extension of patch RPE (Ext. RPE), before stopping at areas of preserved host RPE (Host). Retinal segmentation appears well preserved over the patch.

**Figure 3 diagnostics-14-01005-f003:**
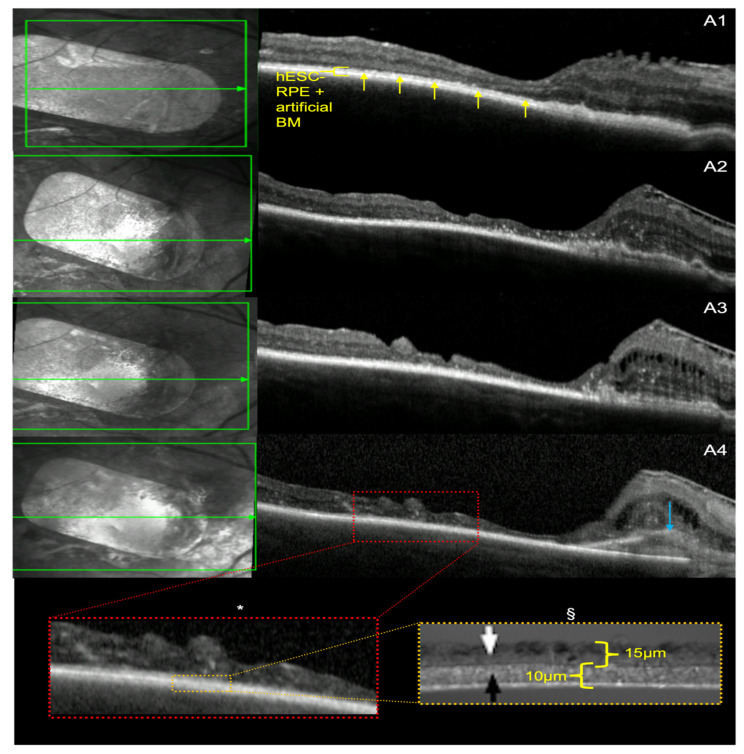
SD-OCT series of subject 1. Near infrared photos at the left part indicate the plane of the b-scan at the right. For subject 1, scans from the same retinal location (**A1**–**A4**) have been exported using the follow-up option of the Heyex software (Heidelberg-Spectralis). (**A1**) SD-OCT from 3-month post-op showing the hESC-RPE and artificial basement membrane bilayer (yellow pointer). Five measurements were taken where the hESC-RPE-BM transplant was uniform in appearance. These were repeated to have 2 measurements per 5 points pre-defined by the yellow arrows. This was done for all further scans (**A2**–**A4**). There is evidence of a nasal epiretinal membrane (ERM) that progressed over time. There is also the development of intraretinal cysts. (**A2**) SD-OCT at year 1 post-op showing ongoing ERM and intraretinal cysts. (**A3**) SD-OCT scans from post-op year 2 showing stabilisation of the retinal thickness over the bulk of the patch and increasing cystic changes in the nasal part of the treated retina. (**A4**) SD-OCT at year 5 post-op, with evidence of RPE thickening progression over the 5 years, highlighted by the blue arrow. The magnified view of the macula (**A4** *) shows apparent hESC-RPE with artificial BM, which corresponds to the histological structure of the hESC-RPE highlighted by the white arrow (**A4** §) which has a thickness of 15 microns [18], and the artificial BM highlighted by the black arrow which has a thickness of 10 microns [5] (**A4** §), with a combined depth of 25 µm.

**Figure 4 diagnostics-14-01005-f004:**
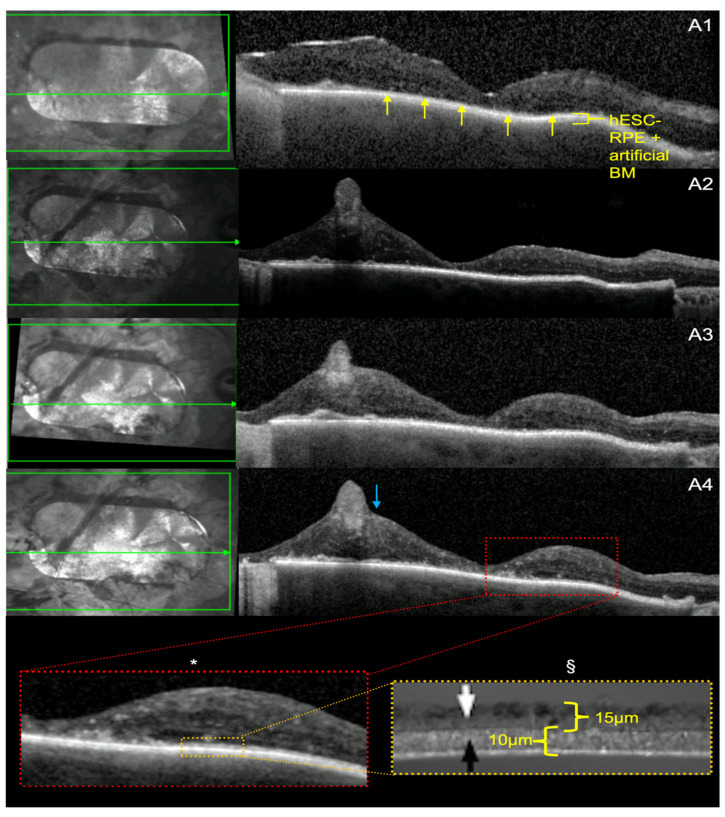
SD-OCT series of subject 2. Near infrared photos at the left part indicate the plane of the b-scan at the right. For subject 2, scans from the same retinal location (**A1**–**A4**) have been exported using the follow-up option of the Heyex software (Heidelberg-Spectralis). (**A1**) SD-OCT from 3 months post-transplantation of hESC RPE-BM transplant, which is highlighted by the yellow pointer. Five measurements were taken where the hESC-RPE-BM transplant was uniform in appearance. These were repeated to have 2 measurements per 5 points pre-defined by the yellow arrows. This was done for all further scans (**A2**–**A4**). There is a temporal detachment of hESC-RPE from the artificial BM as previously described. There is development of intraretinal cysts. (**A2**) SD-OCT scans from post-op year 1, showing stabilisation of the retinal thickness over the bulk of the patch, with some thickening later cystic changes in the nasal part of the treated retina at year 2 (**A3**). (**A4**) SD-OCT scan through the whole patch from post-op year 5 with persistent epiretinal fibrosis as highlighted by the blue arrow. The magnified view of the macula (**A4** *) shows apparent hESC-RPE with artificial BM, which corresponds to the histological structure of the hESC-RPE highlighted by the white arrow (**A4** §) which has a thickness of 15 microns [18], and the artificial BM highlighted by the black arrow which has a thickness of 10 microns [5] (**A4** §), with a combined depth of 25 µm.

**Figure 5 diagnostics-14-01005-f005:**
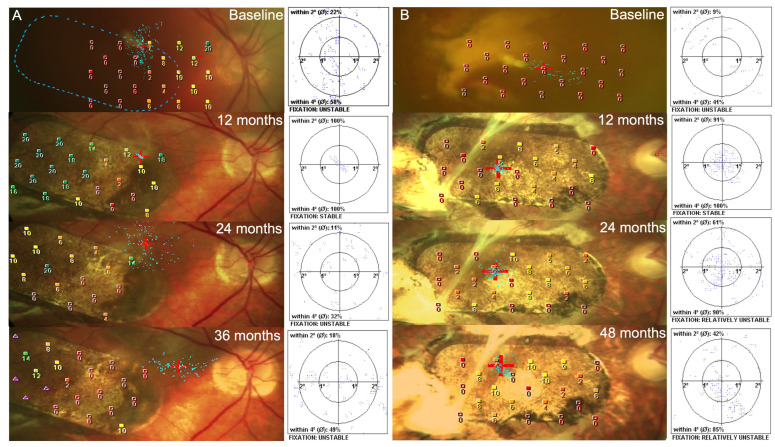
Fixation and microperimetry examinations using the Nidek MP-1 microperimeter for subjects 1 and 2 (**A**,**B**). Colour fundus photos have an overlay of the microperimetry grid with numerical retinal sensitivity (dB) measured at each loci. The cross represents the centre of fixation with the cloud of blue dots statistically identifying the retinal area involved in fixating the target. At baseline, there is limited retinal sensitivity over the macula for either study eye. The blue dotted lined for subject 1 at baseline (**A**, Baseline) represents a pre-operative approximation of the retina treated with the hESC-RPE patch. This was not possible for subject 2 due to the large submacular haemorrhage and poor view on the colour fundus photos. Timepoints for subjects 1 and 2 are from baseline, 12 months, 24 months and 36 months for subject 1 and 48 months for subject 2.

**Table 1 diagnostics-14-01005-t001:** Average hESC-RPE and BM thickness measurements for subject 1 and subject 2. As per Figure 3 and Figure 4, five measurements were taken and repeated once for hESC-RPE and BM thickness for study participant 1 and study participant 2 at 3 months, 12 months, 2 years (24 months) and 5 years (60 months) post-operatively. This table shows the average of 5 readings which were repeated once for each participant at each visit. Intraclass correlation coefficient (ICC) was performed on the raw data (5 measurements repeated once per OCT scan), to determine intra-rater reliability, using a two-way mixed effects model, looking at absolute agreement for a single rater. Confidence intervals are also provided (CIs) for the ICC.

Average hESC-RPE and BM Thickness Measurement		
Time Point	Study Participant 1	Study Participant 2
3 months	23	24
12 months	27	24
24 months	25	25
60 months	27	27
	ICC	0.98 (CI 0.91, 1.0)

**Table 2 diagnostics-14-01005-t002:** Early Diabetic Retinopathy Study (ETDRS) Best Corrected Visual Acuity (BCVA) Letter score assessment during the 5-year follow up of both subjects. Time points from baseline (0), 3 months, 6 months, 9 months, 12 months, 24 months (2 years), 36 months (3 years), 48 months (4 years) and 60 months (5 years) are documented for both subjects (study participants 1 and 2).

ETDRS BCVA (Letter Score)		
Months	Study Participant 1	Study Participant 2
0	10	8
3	9	23
6	33	30
9	32	30
12	39	29
24	26	23
36	15	22
48	11	20
60	2	20

## Data Availability

1 year data for the trial is available in the paper [5]. 5-year data for the visual and other structural outcomes from the trial will become available once published.

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
