# Peer review of "The Fate of RPE Cells Following hESC-RPE Patch Transplantation in Haemorrhagic Wet AMD: Pigmentation, Extension of Pigmentation, Thickness of Transplant, Assessment for Proliferation and Visual Function—A 5 Year-Follow Up"

_diagnostics, 2024, doi:10.3390/diagnostics14101005_

Round 1
Reviewer 1 Report
Comments and Suggestions for Authors
The originality of the study is exceptional. I suggest some major changes.
Abstract:
Line 15: I suggest explaining in line 15 the significance of hESC-RPE, instead of in line 17.
Line 24: I would remove both.
Introduction:
Line 57: no evidence of cell proliferation.
Material and methods:
Line 105: what is TS?
Results:
Line 124: (Figure 1. A2), which enlarged…
Line 133: define PED.
Lines 135 and 141: first instead of 1st.
Lines 157 and 183: Colour fundus.
Lines 160 and 186: time points.
I suggest adding data about the visual acuity during the follow-up and if there were subjective visual improvement.
Discussion:
Line 334: to confirm that the pigmented areas represent RPE
Line 343: what is FFA?
Line 346: I would eliminate and at the end of the line.
Reviewer 2 Report
Comments and Suggestions for Authors
The topic described in this paper is innovative, useful for the clinicians and gives hope for further development of such implants to restore and save vision in patients who have poor vision due to nAMD and have unmet medical need – no ways to cure it.
The introduction is sufficient, the results are presented in a good way, descriptive and also sufficient. Discussion could me more detailed, but in present view gives more or less enough topics covered. Separately it is worth mentioning brilliant surgery: “jewelry work” by the surgeon and repetitive after surgery measurements are very accurate.
These results are very much awaited, confirming relatively safe implantation. Expected complications after implantation over a distant observing period (5 yrs) are discussed: retina thinning, epiretinal membrane, fibrosis, remodeling.
Major concerns:
1. Oncology – plan should be developed and written. Is there any standard procedure of clinical oncologist follow up for the later observation period: how frequently, what particular measurements and scores?
2. Visual acuity – before – after. Visual function assessment results over the 5 years of observation – visual acuity, pericentral visual fields (microperimetry) or other relevant method.
3. More patients involved and analyzed are needed in this study to make more robust conclusions.
Minor comments:
1. Line 42 – centred – correct to: centered
2. Line 57 no evidence cell proliferation -- no evidence of cell proliferation
3. Line 104 - ImageJ2 – space is needed? Double check spelling.
4. Formatting issues: layout before 200 – font is smaller, after line 202 – it’s bigger.
5. In some cases, spaces are lacking, however sometimes there are excessive spaces (i.e. line 217 and others)
6. Table 1 is a picture of poor quality. Should be replaced with actual table.
After minor revision and formatting this paper is highly recommended for publication.
Comments on the Quality of English LanguageMinor editing required.
Round 2
Reviewer 1 Report
Comments and Suggestions for Authors
No comments